# Estimating the Robustness of Classification Models by the Structure of the Learned Feature-Space

[1][2] **Kalun Ho,** [1]**Avraam Chatzimichailidis,** [1]**Franz-Josef Pfreundt,**
[1][4]**Janis Keuper,** [2][3]**Margret Keuper**

[1] Competence Center High Performance Computing, Fraunhofer ITWM, Kaiserslautern
Fraunhofer Research Center Machine Learning, Germany
[2] University of Siegen [3] Max Planck Institute for Informatics, Saarland Informatics Campus, Germany
[4] Institute for Machine Learning and Analytics (IMLA), Offenburg University, Germany
kalun.ho@itwm.fraunhofer.de

## Abstract

Over the last decade, the development of deep image classification networks has mostly been driven by the search for the best performance in terms of classification accuracy on standardized benchmarks like ImageNet. More recently, this focus has been expanded by the notion of model robustness, *i.e.* the generalization abilities of models towards previously unseen changes in the data distribution. While new benchmarks, like ImageNet-C, have been introduced to measure robustness properties, we argue that fixed testsets are only able to capture a small portion of possible data variations and are thus limited and prone to generate new overfitted solutions. To overcome these drawbacks, we suggest to estimate the robustness of a model directly from the structure of its learned feature-space. We introduce robustness indicators which are obtained via unsupervised clustering of latent representations from a trained classifier and show very high correlations to the model performance on corrupted test data.

## 1 Introduction

Deep learning approaches have shown rapid progress on computer vision tasks. Much work has been dedicated to train ever deeper models with improved validation and test accuracies and efficient training schemes (Zoph et al. 2018; Howard et al. 2017; Liu et al. 2018; Hu, Shen, and Sun 2018). Recently, this progress has been accompanied by discussions on the robustness of the resulting model (Djolonga et al. 2020). Specifically, the focus shifted towards the following two questions: 1. How can we train models that are robust with respect to specific kinds of perturbations? 2. How can we assess the robustness of a given model? These two questions represent fundamentally different perspectives on the same problem. While the first question assumes that the expected set of perturbations is known during model training, the second question rather aims at estimating a models behavior in unforeseen cases and predict its robustness without explicitly testing on specific kinds of corrupted data.

In this paper, we address the second research question. We argue that the clustering performance in a model's latent space can be an indicator for a model's robustness.

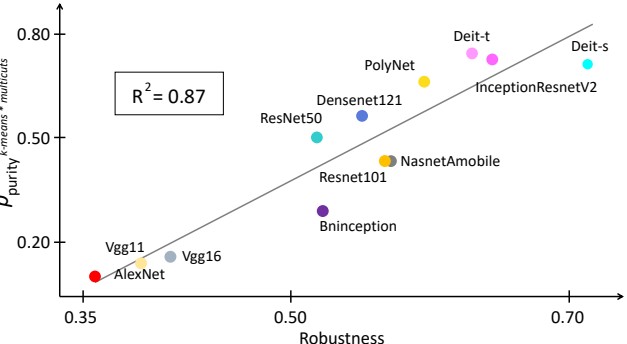

Figure 1: Predicting the robustness of models using our proposed cluster purity indicator ($p_{purity}$): The correlation between $p_{purity}$ of models trained on ImageNet with the measured test accuracy on ImageNet-C is $R^2 = 0.87$.

For this purpose, we introduce cluster purity as a robustness measure in order to predict the behavior of models against data corruption and adversarial attacks. Specifically, we evaluate various classification models (Krizhevsky, Sutskever, and Hinton 2012; Zoph et al. 2018; Huang et al. 2017; He et al. 2016; Szegedy et al. 2017; Zhang et al. 2017; Ioffe and Szegedy 2015; Touvron et al. 2020) on the ImageNet-C (Hendrycks and Dietterich 2019) dataset of corrupted ImageNet images where we measure the robustness of a model as the ratio between the accuracy on corrupted data and clean data. The key result of this paper is illustrated in figure 1: it shows that the model robustness is strongly correlated to the relative clustering performance on the models' latent spaces, i.e. the ratio between the cluster purity and the classification accuracy, both evaluated on clean data. The clusterability of a model's feature space can therefore be considered as an easily accessible indicator for model robustness.

In summary, our work contributes the following:

- We study the feature spaces of several ImageNet pre-trained models including the state-of-the-art CNN models (Zoph et al. 2018; Huang et al. 2017; He et al. 2016; Szegedy et al. 2017; Zhang et al. 2017) and the recently proposed transformer models (Touvron et al. 2020) and evaluate their model robustness on the ImageNet-C dataset and against adversarial attacks.

- We show that intra- and inter-class distances extracted from classification models are not suitable as a direct indicator for a model's robustness.

- We provide a study of two clustering methods, *K-means* and the *Minimum Cost Multicut Problem* (MP) and analyze the correlation between classification accuracy, robustness and clusterability.

- We show that the relative clustering accuracy, *i.e.* the ratio between classification and clustering performance, is a strong indicator for the robustness of the classification model under ImageNet-C corruptions.

This paper is structured as follows: We first review the related work on image classification, model robustness and deep clustering approaches in Section 2, then we propose the methodology for the feature space analysis in Section 3. Our experiments and results are discussed in Section 4.

## 2 Related Work

**Image Classification**. Convolutional neural networks (CNN) have shown great success in computer vision. In particular, from the classification of handwritten characters (LeCun et al. 1998) to images (Krizhevsky, Hinton et al. 2009), CNN-based methods consistently achieve state-of-the-art in various benchmarks. With the introduction of ImageNet (Russakovsky et al. 2015), a dataset with higher resolution images and one thousand diverse classes is available to benchmark the classification accuracy of ever better performing networks (Krizhevsky, Sutskever, and Hinton 2012; Zoph et al. 2018; Huang et al. 2017; He et al. 2016; Szegedy et al. 2017; Zhang et al. 2017), ranging from small and compact network (Howard et al. 2017) to large models (Simonyan and Zisserman 2014) with over 100 millions of parameters.

**Transformers**. Recently, transformer network architectures, which were originally introduced in the area of natural language processing (Vaswani et al. 2017), have been successfully applied to the image classification task (Chen et al. 2020; Dosovitskiy et al. 2020). The performance of transformer networks is competitive despite having no convolutional layers. However, transformer models require long training times and large amounts of data (Dosovitskiy et al. 2020) in order to generalize well. A more efficient approach for training has been proposed in (Touvron et al. 2020), which is based on a teacher-student strategy (distillation). Similarly, (Caron et al. 2021) uses the same strategy on self-supervised tasks.

**Model Robustness**. Convolutional neural networks are susceptible to distribution shifts (Quiñonero-Candela et al. 2009) between train and test data (Ovadia et al. 2019; Geirhos et al. 2018; Hendrycks and Dietterich 2019; Saikia,

Schmid, and Brox 2021). This concerns both visible input domain shifts by for example considering corrupted, noisy or blurred data, as well as imperceptible changes in the input, induced by (Moosavi-Dezfooli, Fawzi, and Frossard 2016; Goodfellow, Shlens, and Szegedy 2014; Kurakin, Goodfellow, and Bengio 2016). These explicitly maximize the error rate of classification models (Szegedy et al. 2013; Biggio and Roli 2018) and thereby reveal model weaknesses. Many methods have been proposed to improve the adversarial robustness by specific training procedures, *e.g.* (Moosavi-Dezfooli, Fawzi, and Frossard 2016; Jakubovitz and Giryes 2018). In contrast, input distribution shifts induced by various kinds of noise as modeled in the ImageNet-C (Hendrycks and Dietterich 2019) dataset mimic the robustness of a model in unconstrained environments, for example under diverse weather conditions. This aspect is crucial if we consider scenarios like autonomous driving, where we want to ensure robust behaviour for example under strong rain. Therefore, we focus on the latter aspect and investigate the behaviour of various pre-trained models under ImageNet-C corruptions but also evaluate the proposed robustness measure on adversarial perturbations (Moosavi-Dezfooli, Fawzi, and Frossard 2016; Jakubovitz and Giryes 2018). While (Jiang et al. 2018) propose a *trust score* instead of its models' confidence score to judge the reliability of the results, (Buzhinsky, Nerinovsky, and Tripakis 2021) introduce a natural way of measuring adversarial robustness, called *latent space performance metrics*. In contrast, (Giraudon et al. 2021) measure robustness using a mean radius approach.

**Clustering.** Clustering approaches, deep clustering approaches in particular, have shown to benefit from well structured feature spaces. Such approaches therefore aim at optimizing the latent representations for example using variational autoencoders or Gaussian mixture model or *K-means* priors (Prasad, Das, and Bhowmick 2020; Xie, Girshick, and Farhadi 2016; Ghasedi Dizaji et al. 2017; Ghasedi et al. 2019; Caron et al. 2018). (Caron et al. 2018) iteratively groups points using *K-means* during the latent space optimization. Conversely, we are investigating the actual feature space learned from image classification tasks using clusterability as a measure for its robustness. Therefore, we apply clustering approaches on pre-trained feature spaces. Further, while the above mentioned methods rely on a *K-means*-like clustering, *i.e.* data is clustered into a given number of clusters, we also evaluate clusters from a similarity driven clustering approach, the *Minimum Cost Multicut Problem* (Bansal, Blum, and Chawla 2004).

The Multicut Problem, aka. Correlation Clustering, groups similar data points together by pairwise terms: data (*e.g.* images) are represented as nodes in a graph. The real valued weight of an edge between two nodes measures their similarity. Clusters are obtained by cutting edges in order to decompose the graph and minimize the cut cost. This problem is known to be NP-hard (Demaine et al. 2006). In practice, heuristic solvers often perform reasonably (Kernighan and Lin 1970; Beier et al. 2014). Correlation Clustering has various applications in computer vision, such as motion tracking and segmentation (Keuper et al. 2018; Wolf et al.

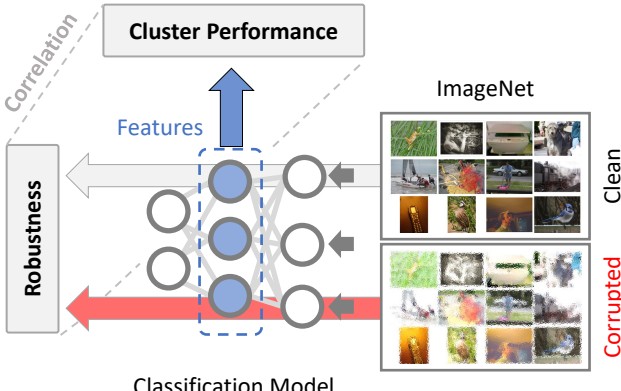

Figure 2: The robustness of a model is measured by its relative classification performance, which is the ratio between clean and corrupted (in red arrow) data.. The latent space or features (in blue) of various classification models is sampled using ImageNet images. The feature representations are then clustered with the *K-means* and *Multicut* clustering approaches. The correlation is visualized in 1.

2020), image clustering (Ho et al. 2020a) or multiple object tracking (Tang et al. 2017; Ho et al. 2020a).

## 3 Feature Space Analysis

Our aim is to establish indicators for a model's robustness from the structure of its induced latent space. Therefore, we first extract latent space samples, *i.e.* feature representations of input test images. The latent space structure is subsequently analyzed using two different clustering approaches. *K-means* is clustering data based on distances to a fixed number of cluster means and can therefore be interpreted as a proxy of how well the latent space distribution can be represented by a univariate Gaussian mixture model. The *Minimum Cost Multicut problem* formulation clusters data points based on their pairwise distances and therefore imposes less constraints on the data manifold to be clustered. Figure 2 gives an overview of the methodology. First, we briefly recap classification models as feature extractors in Section 3.1. The *K-means* and *Minimum Cost Multicut Problem* on the image clustering task are explained in Section 3.2. In Section 3.3, we review evaluation metrics for measuring the clustering performance and in Section 3.4, we present our proposed metrics for robustness estimation.

### 3.1 Extracting Features from Classification Models

Classification models with multiple classes are often trained with softmax cross-entropy and it has been shown that features, learned from vanilla softmax cross-entropy achieve a high performance in transfer accuracy (Kornblith et al. 2020). In order to obtain the learned features from images, the last layer of the trained model (classifier) is removed, which is often done for instance in transfer learning (Sharif Razavian et al. 2014; Shin et al. 2016) or cluster-

Table 1: **Classification models:** all models are trained and evaluated on the ImageNet (Russakovsky et al. 2015) dataset, sorted by performance. We report the Top1 classification accuracy in %. The first ten models are based on convolutional layers while the last two are transformer networks.

| MODEL | FEATURES | PARAM | TOP1 % |
|---|---|---|---|
| ALEXNET | 4096 | 61.1M | 56.4 |
| VGG11 | 4096 | 132.9M | 69.0 |
| VGG16 | 4096 | 138.4M | 71.6 |
| BNINCEPTION | 1024 | 11.3M | 73.5 |
| NASNETAMOBILE | 1056 | 5.3M | 74.1 |
| DENSENET121 | 1024 | 7.9M | 74.6 |
| RESNET50 | 2048 | 25.6M | 76.0 |
| RESNET101 | 2048 | 44.5M | 77.4 |
| INCRESNV2 | 1536 | 55.8M | 80.2 |
| POLYNET | 2048 | 95.3M | 81.0 |
| DEIT-TINY | 192 | 5.9M | 74.5 |
| DEIT-SMALL | 384 | 22.4M | 81.2 |

ing tasks (Xie, Girshick, and Farhadi 2016). The model encodes an image $x_i$ with a function $f_\theta(.)$, with pre-trained parameters $\theta$. Table 1 shows the different classification models with their according feature dimensions as well as the number of parameters and their top 1 classification accuracy in %. We investigate models which vary significantly in their architectures, including CNNs and transformer models, their number of parameters, ranging from 3.5M to 138M, as well as their test accuracy, ranging from 56.4% to 81.2% top-1 scores. We use features extracted from the full ImageNet test set as latent space samples for our analysis as shown in 2.

### 3.2 Latent Space Clustering

**K-means.** *K-means* is a simple and effective method to cluster $N$ data points into $K$ clusters $S_k$, $k = 1, \ldots, K$. As $K$ is set a priori, this method produces exactly the number of defined clusters by minimizing the intra-cluster distance:

$$\sum_{k=1}^{K} \sum_{x_i \in S_k} ||f(x_i) - \mu_k||^2 \qquad (1)$$

where the centroid $\mu_k$ is computed as the mean of features $\frac{1}{|S_k|} \sum_{x_i \in S_k} f(x_i)$ in cluster $k$.

**Multicut Clustering.** The Minimum Cost Multicut Problem is a graph based clustering approach. Considering an undirected graph $G = (V, E)$, with $v \in V$ being the images $x_i$ of the dataset $X$ with $|V| = N$ samples, a complete graph with $N$ nodes has in total $|E| = \frac{N(N-1)}{2}$ edges. A real valued cost $w : E \to \mathbb{R}$ is assigned to every edge $e \in E$. While the decision, whether an edge is joined or cut, is made based on the edge label $y : E \to \{0, 1\}$, the decision boundary can be derived from training parameters of the model (Ho et al. 2020b), directly learned from the dataset (Ho et al. 2020a; Tang et al. 2017) or simply estimated empirically (via pa-

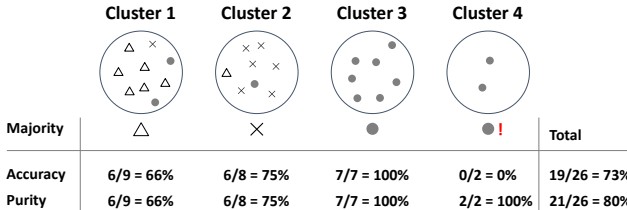

| | Cluster 1 | Cluster 2 | Cluster 3 | Cluster 4 | Total |
|---|---|---|---|---|---|
| Majority | △ | × | ● | ●! | |
| Accuracy | 6/9 = 66% | 6/8 = 75% | 7/7 = 100% | 0/2 = 0% | 19/26 = 73% |
| Purity | 6/9 = 66% | 6/8 = 75% | 7/7 = 100% | 2/2 = 100% | 21/26 = 80% |

Figure 3: Evaluation metrics with 4 clusters with 3 unique classes. **Cluster Accuracy:** The best match for class *dark circle* is cluster 3, since it contains the most frequent items from the same class. Cluster 4 is considered as false positive. **Purity score** on the other hand does not penalize cluster 4. Thus, the purity score is higher than the cluster accuracy (80% vs. 73%).

rameter search). The inference of such edge labels is defined as follows:

$$\min_{y \in \{0,1\}^E} \sum_{e \in E} w_e y_e \qquad (2)$$

$$\text{s.t.} \quad \forall C \in cycles(G) \quad \forall e \in C : y_e \leq \sum_{e' \in C \setminus \{e\}} y_{e'} \qquad (3)$$

Here, edges with negative costs $w_e$ have a high probability to be cut. Equation (3) enforces that for each cycle in $G$, a cut is only allowed if there is at least another edge being cut as well, which was shown in (Chopra and Rao 1993) to be sufficient to enforce on all *chordless* cycles.Practically, the edge costs are computed from pairwise distances in the feature space. The distance $d_{i,j}$ between two features $f(x_i)$ and $f(x_j)$ is calculated from the pre-trained model or encoder $f$, where $x_i$ and $x_j$ are two distinct images from the test dataset, respectively, as

$$d_{i,j} = ||f(x_i) - f(x_j)||^2 . \qquad (4)$$

A logistic regression model estimates the probability $d_{i,j}$ of the edge between $f(x_i)$ and $f(x_j)$ to be cut. This cut probability is then converted into real valued edge costs $w$ using the *logit* function $\text{logit}(p) = \log \frac{p}{1-p}$ such that similar features are connected by an edge with positive, *i.e.* attractive weight and dissimilar features are connected by edges with negative, *i.e.* repulsive weight. The decision boundary (*i.e.* the threshold on $d$, which indicates when to cut or to join) is estimated empirically

### 3.3 Cluster Quality Measures

We use two popular external evaluation metrics (*i.e.* label information are used) to measure the clustering performance: Cluster Accuracy (ACC) and Purity Score. The former metric is calculated based on the Hungarian algorithm (Kuhn 2005), where the best match between the predicted and the true labels are found. The purity score assigns data in a cluster to the class with the most frequent label (Jain, Grover,

and LIET 2017). Formally, given a set of $K$ clusters $S_k$ and a set of classes $L$ with a total number of $N$ data samples, the purity is computed as follows:

$$\frac{1}{N} \sum_{k \in K} \max_{\ell \in L} |S_k \cap \ell| \qquad (5)$$

The advantage of using this metric is two-fold: on one hand, it is suitable if the dataset is balanced and on the other hand, purity score does not penalize having a large number of clusters. Figure 3 depicts an example of both metrics.

### 3.4 Performance Measure

Next, we derive a measure based on the latent space clustering performance, that allows to draw conclusions on a model's robustness without evaluating the model on corrupted data. Thereby, we measure a model's robustness as its relative classification accuracy, *i.e.* the ratio between its classification accuracy on corrupted data and on clean data:

$$Robustness = \frac{\text{Model}_{\text{ACC}^*_{s,c}}}{\text{Model}_{\text{ACC}}} \qquad (6)$$

Parameters $c$ and $s$ are corruption type and severity level (or intensity), respectively for non-adversarial attacks such as ImageNet-C. The aggregated value over all severity levels $s \in \tilde{S}$ on all corruption types $c \in CORR$ is calculated as follows:

$$\text{ACC}^*_{all} = \frac{1}{|\text{CORR}|} \sum_{c \in \text{CORR}} \frac{1}{|\tilde{S}|} \sum_{s=1}^{|\tilde{S}|} \text{ACC}^*_{s,c} \qquad (7)$$

According to equation 6, perfectly robust models therefore have a robustness of 1, smaller values indicate lower robustness. Based on the above considerations on model robustness and clustering performance, we propose to consider the relative clustering performance as an indicator for the model robustness and show empirically that there exists a strong correlation between both. The relative clustering performance, *i.e.* the ratio between clustering performance and classification accuracy $\text{Model}_{\text{ACC}}$ is defined as follows:

$$p = \frac{\text{clustering performance}}{\text{Model}_{\text{ACC}}} \qquad (8)$$

Here, we consider the clustering accuracy $\text{C}_{\text{ACC}}$ and purity score $\text{C}_{\text{purity}}$ as a performance measures for our experiments, *i.e.*

$$p_{\text{ACC}} = \frac{\text{C}_{\text{ACC}}}{\text{Model}_{\text{ACC}}} \quad \text{and} \quad p_{\text{purity}} = \frac{\text{C}_{\text{purity}}}{\text{Model}_{\text{ACC}}}$$

respectively.

**Correlation Metrics.** The degree of correlation is computed based on the coefficient of determination $R^2$ and Kendall rank correlation coefficient $\tau$, respectively with a value of 1.0 being perfectly correlated while 0 means no correlation at all. An example for $R^2$ is illustrated in Figure 1 and $\tau$ in Figure 7.

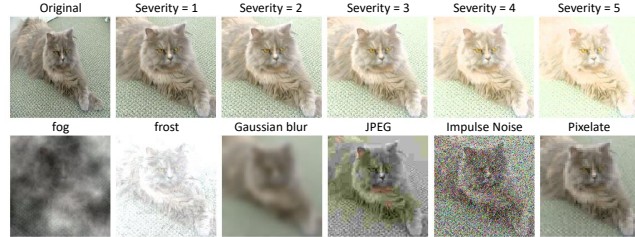

Figure 4: ImageNet-C dataset: first row shows the original image and the corruption *brightness* for different severity levels. Second row: examples of other corruption types at severity level 5.

**Baseline Indicator: Class Overlap $\Delta$.** Our hypothesis is that an initial well-separated feature space of a classification model provides a good estimate regarding the model robustness. A simple method to determine such a separation would be to observe the intra- and inter-class distances between data samples in the feature space. If an overlap between classes exists, they are not well separated, which may indicate weak models. We define this setting as a baseline in order to show that latent space clustering provides significantly more information.

To investigate this, we define the overlap $\Delta$ between the intra- and inter-class distances as follows:

$$\Delta = (\mu_{intra} + \sigma_{intra}) - (\mu_{inter} + \sigma_{inter}) \qquad (9)$$

$\mu$ and $\sigma$ represent the mean and standard deviation of the intra- and inter-class distances.

## 4 Experiments

This section is structured as follows: we first explain the setup of our experiments in 4.1. Then, we present the clustering results in Section 4.2 where we analyse the clustering accuracy and purity for the two considered clustering approaches on the feature spaces of the different models. Section 4.3 shows that the intra- and inter-class distances cannot directly be used as robustness indicators. In Section 4.4, we consider the relationship between the model classification robustness under corruptions and the relative clustering performance of the considered clustering methods and metrics. We show that both clustering accuracy and cluster purity, computed on the feature spaces of clean data, allow to derive indicators for a model's expected robustness under corruptions. Thereby, the purity score is more stable than the clustering accuracy and the information provided by *k-means* clustering and *multicuts* complement one another. In Section 4.5, we evaluate the proposed robustness indicator in the context of adversarial attacks.

### 4.1 Setup

Our experiments are based on the ImageNet (Russakovsky et al. 2015) dataset. All models were pre-trained on the original training dataset. We evaluate 10 CNN-based models and 2 transformer architectures *deit-t* and *deit-s* (t stands for tiny

and s for small). An overview is provided in Table 1. We evaluate the robustness of the considered models against corruptions using the *ImageNet-C* (Hendrycks and Dietterich 2019) dataset and report the model accuracy for classification and clustering tasks. Figure 4 illustrates an example of considered image corruptions: the first row shows the different severity levels $s = 1, \ldots, 5$ of the corruption *brightness*, with 1 being the lowest and 5 the strongest corruption. The second row shows other kinds of image perturbations $c$ at severity level 5 such as *fog*, *frost*, *Gaussian blur*, *jpeg_compression* or *pixelate*. Each corruption $c$ has 5 severity levels $s = 1, \ldots, 5$. All models are trained on the clean dataset and the numbers are evaluated on the full test dataset, as done in (Hendrycks and Dietterich 2019).

### 4.2 Classification vs. Clustering

Table 2 summarizes the evaluation in three categories: classification, *K-means* and *Multicuts*. There are in total $|\text{CORR}| = 19$ corruption types with each $|\tilde{S}| = 5$ severity levels on ImageNet-C. For the classification task, the numbers are reported in top 1% accuracy for all five levels of corruption (denoted as $1 - 5$). On *K-means* and *Multicuts*, we report the clustering metrics as presented in 3.3.

The transformer *deit-s* shows the highest top 1% accuracy on the classification task both on clean and on corrupted data for all severity levels. *Inceptionresnetv2* and *polynet* perform only slightly worse on clean data but are more strongly affected by the ImageNet-C data corruptions than *deit-s*. *Alexnet* shows the worse performance across all corruption levels. Although *resnet50* outperforms *bninception*, *nasnetamobile* and *densenet121* it is less robust against corruption. This is also illustrated in Figure 6 (right).

Considering the clustering accuracy and purity, the *K-means* and the *Multicut* behave significantly different from one another. *K-means* clustering achieves about 70% accuracy for models with the highest clean classification accuracy. Yet, its accuracy is much better for the *deit-t* latent space than for example for the *densenet121* induced latent space, although the clean classification accuracy of both networks is comparable. Overall, the *K-means* clustering works surprisingly well on the transformer models. The *Multicut* clustering showed the highest clustering accuracy on the *inceptionresnetv2* model. The cluster purity was comparably high for the best transformer model *deit-s*. Note that our goal is to derive from the clustering performance an indicator for model robustness, *i.e.* we expect clustering to be less accurate when models are less robust to noise.

### 4.3 Baseline Indicators: Intra- and Inter class-distances

Table 3 shows the correlation (as $R^2$) and the ranking correlation $\tau$ between the class overlap baseline indicator $\Delta$, which we detailed in section 3.3, and the model robustness, grouped by severity level. We use equation 6 to calculate the robustness for severity level $s$ over all corruptions and compare them with $\Delta$. The last column shows the correlation on all corruption levels. All 12 models are considered. The rank correlation $\tau$ is calculated by comparing the

Table 2: Evaluation of robustness on classification and clustering tasks with the *ImageNet C* dataset, evaluated on corruption severity levels 1 to 5. Column *CLEAN* represents the classification performance of the models on the clean dataset. Columns 1-5 show the classification accuracy (top 1%) under different severity levels of over all 19 corruptions and column $\text{ACC}^*_{all}$ shows the mean over corruptions on all 5 severity levels. On *K-means*, *ACC* and *purity* are clustering performance on clean test data. The numbers on *multicuts* are evaluated on a subset. The best score on each column is marked in **bold**.

| | CLASSIFCATION ACCURAY (TOP 1%) | | | | | | | K-means | | | MULTICUTS | | |
| --- | --- | --- | --- | --- | --- | --- | --- | --- | --- | --- | --- | --- | --- |
| MODEL | CLEAN | 1 | 2 | 3 | 4 | 5 | $\text{ACC}^*_{all}$ | ACC | PURITY | $\text{ACC}^*_{all}$ | ACC | PURITY | $\text{ACC}^*_{all}$ |
| ALEXNET | 56.4 | 35.9 | 25.4 | 18.9 | 12.7 | 8.0 | 20.2 | 14.6 | 18.4 | 8.0 | 8.0 | 28.1 | 2.6 |
| VGG11 | 69.0 | 47.3 | 35.3 | 25.7 | 16.7 | 10.1 | 27.0 | 28.0 | 32.8 | 12.4 | 15.8 | 27.2 | 2.5 |
| VGG16 | 71.6 | 50.9 | 38.6 | 28.5 | 18.7 | 11.4 | 29.6 | 32.3 | 37.5 | 14.4 | 19.3 | 27.6 | 2.8 |
| BNINCEPTION | 73.5 | 59.4 | 48.4 | 38.8 | 27.2 | 17.7 | 38.3 | 40.5 | 44.0 | 18.0 | 11.5 | 46.6 | 7.9 |
| NASNETAMOBILE | 74.1 | 60.7 | 51.3 | 43.7 | 33.6 | 22.5 | 42.4 | 41.0 | 45.3 | 23.6 | 41.9 | 70.2 | 19.3 |
| DENSENET121 | 74.6 | 60.2 | 50.9 | 42.2 | 31.4 | 21.0 | 41.1 | 48.9 | 52.1 | 23.4 | 16.8 | 80.5 | 8.3 |
| RESNET50 | 76.0 | 60.1 | 49.8 | 40.1 | 28.9 | 18.8 | 39.6 | 55.8 | 58.6 | 24.9 | 29.3 | 64.3 | 11.1 |
| RESNET101 | 77.4 | 63.6 | 54.4 | 45.5 | 34.0 | 22.9 | 44.1 | 59.1 | 61.9 | 29.3 | 28.6 | 53.7 | 16.6 |
| INCEPTIONRESNETV2 | 80.2 | 68.8 | 60.8 | 53.5 | 43.5 | 31.6 | 51.7 | **70.0** | **71.2** | 37.4 | **71.3** | **81.3** | **39.8** |
| POLYNET | 81.0 | 68.0 | 58.9 | 49.9 | 38.2 | 26.3 | 48.3 | 67.8 | 69.7 | 34.8 | 54.4 | 76.6 | 24.1 |
| DEIT-T | 74.5 | 63.3 | 55.9 | 48.7 | 38.9 | 28.1 | 47.0 | 57.4 | 60.0 | 31.7 | 33.0 | 91.9 | 19.5 |
| DEIT-S | **81.2** | **72.1** | **66.1** | **60.2** | **51.3** | **39.7** | **57.9** | 68.8 | 70.8 | **43.4** | 49.4 | 81.1 | 29.6 |

Table 3: **Baseline** indicators for model robustness: The table shows the correlation between overlap $\Delta$ and model robustness for different corruption severity levels. Second row shows the rank correlation $\tau$ between the actual model robustness rank and the predicted rank using $\Delta$.

| | SEVERITY | | | | | TOTAL |
| --- | --- | --- | --- | --- | --- | --- |
| METRIC | 1 | 2 | 3 | 4 | 5 | |
| $R^2$ | 0.27 | 0.29 | 0.27 | 0.25 | 0.26 | 0.27 |
| $\tau$ | 0.48 | 0.52 | 0.52 | 0.52 | 0.52 | 0.48 |

model's robustness rank and the overlap $\Delta$ ranking. Initial well-separated feature spaces (thus a low $\Delta$) should have a high correlation with their model's robustness. Despite its simplicity, this metric $\Delta$ correlates poorly with a highest score of $R^2 = 0.29$ and $\tau = 0.52$. This observation rejects the simple hypothesis about the overlap of intra- and inter-class distances and it suggests that using $\Delta$ is not sufficiently informative as an indicator for model robustness.

### 4.4 Robustness Indicators: Clustering Measures

In the following we evaluate our proposed clustering driven robustness indicator. Specifically, we want to investigate the effects of different clustering measures on the correlation coefficient $R^2$. Table 4 gives an overview of the strength of correlation on different severity levels and clustering metrics on *K-means* and *Multicuts*. Column $\Delta$ shows the correlation on robustness using the overlap of intra- and inter-class distances as previously discussed. Furthermore, the columns *ACC* and *P* are showing the correlation between the model robustness and the clustering accuracy and purity, respectively. The last column shows the combination of both clus-

tering methods in one metric. *K-means* and *Multicuts* have an $R^2$ value of $R^2 = 0.83$ and $R^2 = 0.55$ for clustering accuracy on all corruption levels. On the purity score, both methods show a slightly higher correlation of $R^2 = 0.83$ and $R^2 = 0.71$, respectively for the sum over all corruptions (last row of Table 4). This indicated that latent space clusterability of clean test images K-means is a valid indicator for model robustness under corruptions. However, we show that both clustering methods are complementary when combining their purity scores with

$$p_{\text{purity}^{k-means \cdot multicuts}} = \frac{\text{C}^{k-means}_{\text{purity}} \cdot \text{C}^{multicut}_{\text{purity}}}{\text{Model}_{\text{ACC}}}. \quad (10)$$

This measure shows the highest correlation with the model robustness with $R^2 = 0.87$ (see fig. 1 for the full correlation plot). Additionally, the combination of purity scores of both methods also yields more consistent results across different severity levels.

**Model Ranking.** Next, we evaluate whether our proposed robustness indicator is able to retrieve the correct ranking in terms of model robustness for our set of classification models. The rank correlation is measured as the Kendall rank coefficient $\tau$. Table 5 shows the results for different setups. Here, *K-means* shows a more consistent and better correlation with highest rank correlation of $\tau = 0.82$ on *ACC* and *Purity*. Again, all clustering metrics outperform the $\Delta$ baseline. Figure 6 illustrates one example of the change of rank between the predicted (left) and actual (right) model robustness. The prediction is done using $p_{\text{ACC}^{K-means}}$, which has a rank correlation of $\tau = 0.79$. Our proposed measure is able to rank different models according to their robustness. The three worse performing models (*alexnet*, *vgg11* and *vgg16*) are correctly retrieved. The largest ranking gap of 3 positions

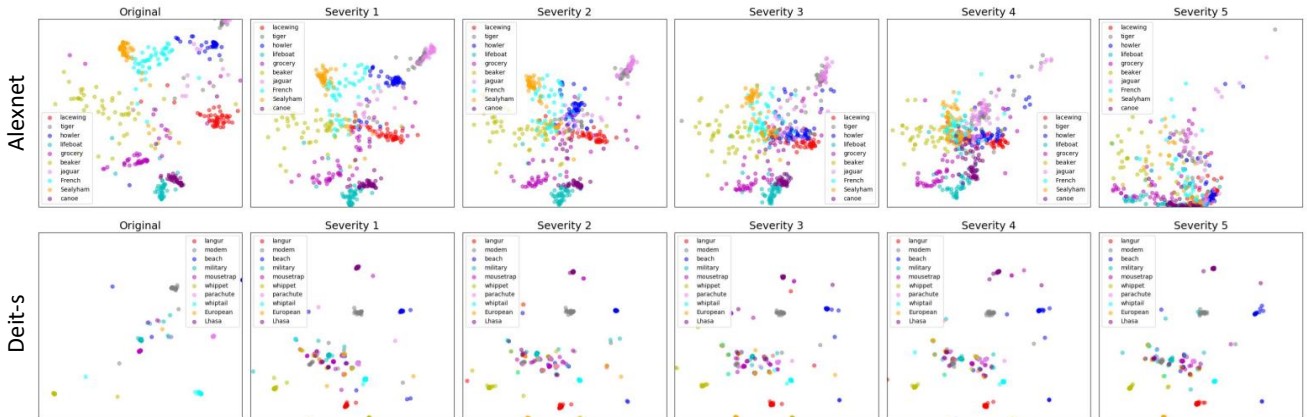

Figure 5: Visualization of feature space on *alexnet* and *deit-s* using umap. The colors correspond to the class labels, where only 10 classes were selected at random. First column shows the initial, clean latent space from the classification model. Each new column depicts the corresponding severity level of the corruption *brightness*. While *alexnet* collapses as the severity increases, the most robust model *deit-s* preserves the clusters very well even after significant corruptions and thus our proposed clusterability of latent space provides a good indicator about the model robustness.

Table 4: **Correlation with different metrics and severity levels:** the reported numbers are the coefficient of determination ($R^2$) on different clustering metrics. Column $\Delta$ is the overlap (from Table 3). Column *ACC* and *Purity* (denoted as *P.*) are used to compute the correlation coefficient $R^2$. The last column is the combination of both clustering methods, *i.e.* last column $Purity$ is equation 10. The highest score is marked in bold.

| METRIC: $R^2$ | | K-MEANS | | MULTICUTS | | COMBINED | |
|---|---|---|---|---|---|---|---|
| SEVERITY | $\Delta$ | ACC | P. | ACC | P. | ACC | P. |
| 1 | 0.27 | **0.85** | **0.85** | 0.48 | 0.67 | 0.54 | 0.82 |
| 2 | 0.29 | **0.87** | **0.87** | 0.51 | 0.70 | 0.58 | 0.86 |
| 3 | 0.27 | 0.84 | 0.83 | 0.55 | 0.73 | 0.61 | **0.87** |
| 4 | 0.25 | 0.79 | 0.79 | 0.58 | 0.72 | 0.64 | **0.87** |
| 5 | 0.26 | 0.75 | **0.84** | 0.57 | 0.68 | 0.64 | **0.84** |
| ALL | 0.27 | 0.83 | 0.83 | 0.55 | 0.71 | 0.62 | **0.87** |

| Predicted | | | | Actual | | |
|---|---|---|---|---|---|---|
| $100 \times p_{ACC^{k-means}}$ | Model Name | Rank | | Rank | Model Name | $100 \times$ Robustness |
| 25.9 | AlexNet | 12 | | 12 | AlexNet | 35.8 |
| 40.6 | Vgg11 | 11 | | 11 | Vgg11 | 39.1 |
| 45.1 | Vgg16 | 10 | | 10 | Vgg16 | 41.3 |
| 55.1 | BNInception | 9 | | 9 | Resnet50 | 52.1 |
| 55.3 | NasnetAmobile | 8 | | 8 | BNInception | 52.1 |
| 65.5 | Densenet121 | 7 | | 7 | Densenet121 | 55.1 |
| 73.4 | Resnet50 | 6 | | 6 | Resnet101 | 57.0 |
| 76.4 | Resnet101 | 5 | | 5 | NasnetAmobile | 57.2 |
| 77.0 | Deit-t | 4 | | 4 | PolyNet | 59.6 |
| 83.7 | PolyNet | 3 | | 3 | Deit-t | 63.1 |
| 84.7 | Deit-s | 2 | | 2 | InceptionResnetV2 | 64.5 |
| 87.3 | InceptionResnetV2 | 1 | | 1 | Deit-s | 71.3 |

Figure 6: Change in robustness ranking based on predicted (left) vs. actual (right) model robustness on ImageNet-C using clustering metric $p_{ACC^{K-means}}$ on total corruptions $\tau = 0.79$. Top is the least robust model ($Rank = 12$) while the bottom shows the most robust model ($Rank = 1$). The highest score is marked in bold.

is observed for *nasnetamobile* and *resnet50*. In this particular example, the value for *alexnet* is calculated as follows: $\frac{14.6}{56.4} * 100 = 25.9$ and $\frac{20.2}{56.4} * 100 = 35.8$ for predicted and actual values, respectively.

**Latent Space Visualization.** Umap (McInnes, Healy, and Melville 2018), a scalable dimensionality reduction method similar to the popular technique TSNE(Van der Maaten and Hinton 2008), has been applied to features on 10 randomly selected classes of the ImageNet dataset for visualization. Figure 5 shows one example of the corruption *brightness* for 2 different models: the first column shows features without any corruptions (clean). As the severity level increases, a

collapse is observed for instance on *alexnet*: well-separated clusters (*i.e.* different colors) are being pulled into a direction in the latent space as the severity increases. The model with the highest robustness, *i.e. deit-s*, preserves the clusters well, which explains the high relative clustering performance. This verifies our assumption on the correlation between clusterability and robustness of classification models, that were evaluated in ImageNet-C dataset.

## 4.5 Adversarial Robustness

So far, we have shown that our proposed approach can effectively indicate the robustness of classification models to-

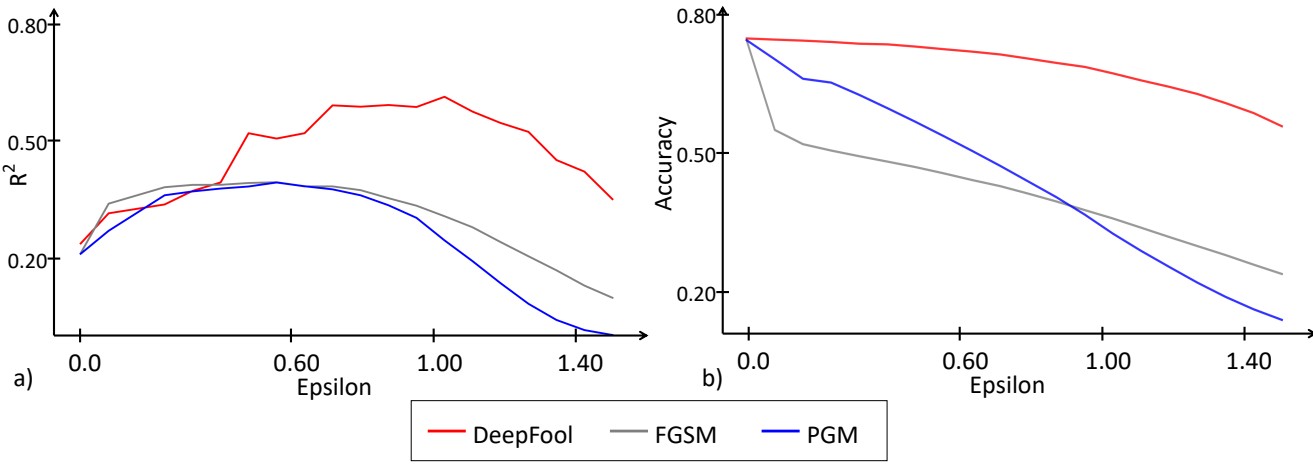

Figure 7: Correlation of our proposed clustering metric on different adversarial attacks with different strengths (*Epsilon*). Left (a): line represents the coefficient of determination $R^2$. Right (b) shows the clustering accuracy. The higher the strength, the weaker the performance.

Table 5: **Rank correlation with different metrics and severity levels:** the reported numbers are the coefficient of determination ($\tau$) on different clustering metrics. Column $\Delta$ is the overlap (from Table 3). Column *ACC* and *Purity* (denoted as *P.*) are the used compute the rank correlation coefficient $\tau$. The last column is the combination of both clustering methods, *i.e.* last column $Purity$ is equation 10. The highest score is marked in bold.

| METRIC: $\tau$ | | K-MEANS | | MULTICUTS | | COMBINED | |
|---|---|---|---|---|---|---|---|
| SEVERITY | $\Delta$ | ACC | P. | ACC | P. | ACC | P. |
| 1 | 0.48 | **0.79** | **0.79** | 0.61 | 0.52 | 0.73 | 0.73 |
| 2 | 0.52 | **0.82** | **0.82** | 0.64 | 0.55 | 0.76 | 0.76 |
| 3 | 0.52 | **0.82** | **0.82** | 0.70 | 0.61 | **0.82** | 0.76 |
| 4 | 0.52 | **0.82** | **0.82** | 0.70 | 0.61 | **0.82** | 0.76 |
| 5 | 0.52 | **0.82** | **0.82** | 0.70 | 0.61 | **0.82** | 0.76 |
| ALL | 0.48 | **0.79** | **0.79** | 0.67 | 0.58 | **0.79** | 0.73 |

wards visible image corruptions and shifts in the data distributions provided by the ImageNet-C benchmark. Here, we extend this evaluation to intentional, non-visible corruptions induced by adversarial attacks. Using the proposed clustering metric $p_{\mathrm{purity}^{k-means\cdot multicuts}}$ as an estimator, we evaluate all 12 models with ImageNet test dataset under two adversarial attacks: *DeepFool* (Moosavi-Dezfooli, Fawzi, and Frossard 2016), *FGSM* (Goodfellow, Shlens, and Szegedy 2014) and *PGM* (Kurakin, Goodfellow, and Bengio 2016) with different perturbation sizes *epsilon*. Figure 7 shows the results of both attacks across all 12 models: left (a) represents the correlation of determination $R^2$ while on the right (b) the classification accuracy, respectively. *Epsilon* (x-axis) is the perturbation size of the attacks. For small epsilon, we expect lower correlations since the model accuracy should hardly be affected. As epsilon increases, some models are more robust than others, *i.e.* better preserve their classifica-

tion accuracy. In this range, we see a relatively strong correlation of the proposed indicator and the relative robust accuracy, albeit weaker than the correlation with robustness to corruptions, with $R^2 = 0.66$, $R^2 = 0.44$ and $R^2 = 0.44$ for *DeepFool* and *FGSM* and *PGM*, respectively. When epsilon becomes too large, the correlation becomes weaker. Our method therefore works well for adversarial examples within a certain range of epsilons. In contrast, no gradients need to be computed (e.g. clustering with *Kmeans*), thus requiring less compute resources as opposed to *FGSM* and *PGM*.

## 5 Conclusion

In this work, we presented a study of the feature space of several pre-trained models on ImageNet including state-of-the-art CNN models and the recently proposed transformer models and we evaluated the robustness on ImageNet-C dataset and extended our evaluation on adversarial robustness as well. We propose a novel way to estimate the robustness behavior of trained models by analyzing the learned feature-space structure. Specifically, we presented a comprehensive study of two clustering methods, *K-means* and the *Minimum Cost Multicut Problem* on ImageNet, where the classification accuracy, clusterability and robustness are analyzed. We show that the relative clustering performance gives a strong indication regarding the model's robustness. Both considered clustering methods show complementary behaviour in our analysis: the coefficient of determination is $R^2 = 0.87$ when combining the purity scores of both methods. Our experiments also show that this indicator is lower, albeit still significant for adversarial robustness ($R^2 = 0.66$ and $R^2 = 0.44$). Additionally, our proposed method is able estimate the order of robust models ($\tau = 0.79$) on ImageNet-C. This novel method is simple yet effective and allows the estimation of robustness of any given classification model without explicitly testing on any specific test data. To the best of our knowledge, we are the first to propose such technique for estimating model robustness.

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
