# OpenReview forum: "Estimating the Robustness of Classification Models by the Structure of the Learned Feature-Space"
_AAAI.org/2022/Workshop/AdvML — AAAI-22 AdvML Workshop LongPaper_

### Official Review · Reviewer_vMwH · 2021-11-26
**A new perspective for assessing robustness from the clustering performance of the model's latent space.**

**Rating:** 6
**Confidence:** 4

**Review:**

**Pros:**
Generally this paper proposes a new indicator for the model's robustness based on the clustering performance of the model's latent space. Also, vast experiments demonstrate the correlation between the indicator and the model's robustness under corrupted inputs, and proposed indicator clearly overperforms the naive baseline, i.e. the class overlap in the latent space.

**Cons:**
However, some doubts still need to be addressed:
- It seems that the clustering performance highly depends on latent space samples, so one question is: will the robustness indicator still work when the latent space samples are derived from a different dataset other than the training dataset?
- Considering the adversarial robustness, acquiring adversarial examples of a known model using methods like FGSM or PGD is neither complex nor time-consuming. Therefore, I don't quite understand why we need to explore the latent space features from the samples with clustering, it seems to require more time and computation resources.

---

### Official Review · Reviewer_YBeh · 2021-11-30
**An instresting work.**

**Rating:** 6
**Confidence:** 3

**Review:**

This paper studies robustness indicators of deep models with the goal of better estimating robustness on "unknown" datasets.  The authors argue that fixed test sets (e.g., ImageNet-C) are only able to capture a small portion of possible data variations and are limited and prone to generate new overfitted solutions.  Towards this end, the authors proposed a novel method to estimate the robustness behaviour of trained models by analyzing the learned feature-space structure.

What I really like about this work is taking an empirical approach to understanding the robustness of the models' inner feature space and providing experimental observations.  Overall this paper is generally well written and focuses on an important direction of understanding the models' robustness on unknown datasets.

## Strength
1). The overall presentation of the paper is clear and easy to follow.

2). In addition to the empirical experiments, the paper also provides some theoretical analysis.

3). Interesting intuition around the concept of robustness.

## Weakness
1). Motivation for using clustering methods as a robustness indicator is not well explained.

2). More datasets should be incorporated.  ImageNet-P perturbations are not included in the paper.  Another important recent benchmark not mentioned in the paper is ImageNet-A ). If the authors want to make their claims more reliable, I would encourage them to consider these datasets (in addition to ImageNet and ImageNet-C).

3). Some experimental results are not explained. For example, in Figure 7,  when epsilon increases,  the correlation becomes weaker.  This seems to be a conflict against the main contribution of this method.

---

### Decision · Program_Chairs · 2021-12-01

**Decision:**

Accept (Long Paper)

**Comment:**

Both reviewers give positive ratings on this paper. Please address their concerns in the final version.